# Antibiotic Resistance/Susceptibility Profiles of *Staphylococcus equorum* Strains from Cheese, and Genome Analysis for Antibiotic Resistance Genes

**DOI:** 10.3390/ijms241411657

**Published:** 2023-07-19

**Authors:** Lucía Vázquez, Mariela E. Srednik, Javier Rodríguez, Ana Belén Flórez, Baltasar Mayo

**Affiliations:** 1Departamento de Microbiología y Bioquímica, Instituto de Productos Lácteos de Asturias (IPLA), Consejo Superior de Investigaciones Científicas (CSIC), Paseo Río Linares s/n, 33300 Villaviciosa, Spain; lucia.vazquez@ipla.csic.es (L.V.); javier.rodriguez@ipla.csic.es (J.R.); abflorez@ipla.csic.es (A.B.F.); 2Instituto de Investigación Sanitaria del Principado de Asturias (ISPA), Avenida de Roma s/n, 33011 Oviedo, Spain; 3Department of Veterinary Microbiology and Preventive Medicine, College of Veterinary Medicine, Iowa State University, Ames, IA 50011, USA; mariela.srednik@gmail.com

**Keywords:** *Staphylococcus equorum*, antibiotic resistance, dairy microbiology, starters, adjunct cultures, cheese

## Abstract

In food, bacteria carrying antibiotic resistance genes could play a prominent role in the spread of resistance. *Staphylococcus equorum* populations can become large in a number of fermented foods, yet the antibiotic resistance properties of this species have been little studied. In this work, the resistance/susceptibility (R/S) profile of *S. equorum* strains (*n* = 30) from cheese to 16 antibiotics was determined by broth microdilution. The minimum inhibitory concentration (MIC) for all antibiotics was low in most strains, although higher MICs compatible with acquired genes were also noted. Genome analysis of 13 strains showed the *S. equorum* resistome to be composed of intrinsic mechanisms, acquired mutations, and acquired genes. As such, a plasmidic *cat* gene providing resistance to chloramphenicol was found in one strain; this was able to provide resistance to *Staphylococcus aureus* after electroporation. An *msr*(A) polymorphic gene was identified in five strains. The Mrs(A) variants were associated with variable resistance to erythromycin. However, the genetic data did not always correlate with the phenotype. As such, all strains harbored a polymorphic *fosB*/*fosD* gene, although only one acquired copy was associated with strong resistance to fosfomycin. Similarly, a plasmid-associated *blaR1-blaZI* operon encoding a penicillinase system was identified in five ampicillin- and penicillin G-susceptible strains. Identified genes not associated with phenotypic resistance further included *mph*(C) in two strains and *norA* in all strains. The antibiotic R/S status and gene content of *S. equorum* strains intended to be employed in food systems should be carefully determined.

## 1. Introduction

The discovery of antibiotics is among the 20th century’s most important achievements in human and veterinary medicine [1]. The initial success, however, has been tarnished by the appearance of resistance and the subsequent transfer of the associated genetic determinants to pathogenic bacteria, hindering the treatment of infections [2]. Although the presence of antibiotic resistance in beneficial and commensal bacteria poses no direct risk to human or animal health, populations of such microorganisms are potential reservoirs of resistance genes that pathogens may acquire [3]. The food chain is pivotal in the transmission of such genes, a consequence of the high cell densities and stress conditions the associated microbiota may experience [4]. Transfer can occur either during food processing or intestinal transit [5].

Milk and fermented dairy products are key players in the transmission of antibiotic resistance in food [6]. Large gene loads for antimicrobial resistance have been repeatedly reported in milk and dairy products [7,8]. Interest in antibiotic resistance in the dairy setting has largely focused on lactic acid bacteria (LAB) [9,10], but this clashes with more recent postulates of the One Health concept, which understands all environments of the food chain—and thus their microbiotas—to be connected [11]. Given the huge microbial diversity seen in some cheeses [12], plus the fact that the population sizes of several taxa may exceed those of LAB [13,14], focusing antibiotic resistance only on the latter type of microorganisms is insufficient. However, distinguishing intrinsic from acquired resistance in food-associated bacteria can be challenging, as it is distinguishing acquired resistance due to mutations from resistance due to acquired genes [15]; as regards transference, only the latter resistance is of great concern [15]. Indeed, resistance/susceptibility (R/S) cut-offs for many species have yet to be established [16].

*Staphylococcus equorum* is such a species for which R/S cut-offs have yet to be developed. Indeed, work on antibiotic resistance in *S. equorum* strains has been particularly scant, and involved mostly disk diffusion assays [17,18,19,20,21], which do not allow the establishment of reliable cut-offs. *S. equorum* is a member of the coagulase-negative staphylococci (CoNS), a bacterial group whose components are frequently detected in food-processing environments and fermented foods [22,23]; certainly, *S. equorum* is a majority commensal population in some cheeses [24,25]. Enzyme activities of CoNS species contribute to the development of flavors and other sensorial characteristics of fermented foods [26]. They can also inhibit the growth of undesirable microorganisms [27]. Therefore, although they do not enjoy Generally Regarded as Safe (GRAS) status, and carry no European Qualified Presumption of Safety (QPS) label, selected strains of *S. equorum* have been broadly proposed as starters or adjunct cultures for some fermented foods [28,29,30,31]. No incidents of food poisoning or infections with *S. equorum* (or any other CoNS species) have ever been reported [32]. However, in order not to spread antibiotic resistance genes through the food chain, starters, and adjunct cultures should contain no transferable ARGs.

The present work reports a broth microdilution survey of the MIC values for 16 antibiotics in *S. equorum* strains (*n* = 30) isolated from traditional blue-veined cheeses made from raw milk. To examine the links between strain phenotype and genetics, 13 strains were then subjected to genome sequencing and analysis. This prompted further phenotypic testing with respect to additional antibiotics, such as lincomycin and fosfomycin. Genome analysis also allowed acquired resistance caused by mutations to be distinguished from resistance afforded by acquired genes.

## 2. Results

### 2.1. Antimicrobial Testing

The genotyping results revealed wide genetic diversity among the studied *S. equorum* strains. Indeed, among the 53 isolates, 30 different strains were identified with <90% similarity (i.e., below the repeatability level of the techniques used) (Appendix A). Representative strains of the 30 different typing profiles were then tested for antibiotic resistance by broth microdilution.

The antibiotic R/S survey of *S. equorum* strains returned no intrinsic resistance to the set of antibiotics tested, except for a moderate resistance to chloramphenicol (range 4–16 µg mL^−1^; mode 8 µg mL^−1^) (Table 1).

The majority of the antibiotics showed very low MICs in most strains, suggesting the latter to be highly susceptible. For gentamicin, kanamycin, and neomycin, the MICs in all strains were equal to or lower than the lowest antibiotic concentrations assayed. Low MIC values, just one or two dilutions above the weakest concentrations tested, were also obtained for streptomycin, vancomycin, and ciprofloxacin. Since *S. equorum*-specific cut-offs are yet to be established, the breakpoints proposed by EUCAST and CLSI for CoNS or other *Staphylococcus* species were used (Table 1). These breakpoints strongly suggested that all the *S. equorum* strains were susceptible to the above six antibiotics. The MICs for the remaining antibiotics varied more widely between the strains. Nonetheless, the MICs for tetracycline, linezolid, ampicillin, and trimethoprim were still low and, when available, below the established cut-offs; consequently, the strains were also considered susceptible. For clindamycin, quinupristin-dalfopristin, and rifampicin, the MICs were slightly above the EUCAST clinical breakpoints, but still below those of the CLSI for some strains; these strains were also deemed susceptible. The MICs of penicillin G were above the CLSI cut-off in seven strains (ranging from 0.5 to 16 µg mL^−1^). The same resistance range was obtained by an independent penicillin G microdilution test. However, variability in the strains giving lower or higher MICs was noted. Penicillin testing by an MTS assay proved the seven strains to resist higher antibiotic concentrations than all others, although the MICs were much lower than those obtained with the Sensititre plates (Table 1). In addition, the MTS assay showed the occasional presence of colonies growing on the verge of the inhibition halo, suggesting the presence of low-resistant mutants. Finally, clear-cut MICs compatible with acquired resistance were scored in only four *S. equorum* strains: for erythromycin in 8A3C, 16A1C, and 50A2C (MICs 64, 32, and 24 μg mL^−1^, respectively), and for chloramphenicol in 5A3C (MIC 64 μg mL^−1^) (Appendix A).

The distribution curves of the MICs obtained with ECOFFinder and NRI worksheet programs were almost identical with the two programs for most antibiotics or differed by just one dilution. For the sake of simplicity, only those obtained in the NRI analysis are reported (Appendix A). Narrow-peaked curves were seen for most antibiotics, while the majority of the remainder had normal distribution curves, except for those of ampicillin and penicillin G. Despite this, tentative *S. equorum*-specific R/S cut-offs could still be proposed for all tested antibiotics (Table 1).

### 2.2. Genome Sequencing

Thirteen strains phenotypically either susceptible to all the tested antibiotics or resistant to any of them were selected for genome sequencing. On average, the recorded genome sequences contained 2,937,190 bp and had a GC content of 32.86%. Genome sequences were distributed in a number of contigs, ranging from 24 (in *S. equorum* 23A3C) to 107 (in *S. equorum* 50A2C). Pairwise digital DNA–DNA hybridization (dDDH) of the genome of these strains with those of selected type strains for *Staphylococcus* species returned an identity value >95% to *S. equorum* subsp. *equorum* NCTC 12414^T^ for seven strains (T17, 2A3C, 8A3C, 16A1C, 23A3C, 35A3C, and 50A2C). Three other strains (CL10P, 48A3I, and 1BCExtra) showed the strongest identity to *S. equorum* subsp. *linens* DSM 15097^T^. These results confirmed their previous identification based on the partial amplification and analysis of their 16S rRNA genes, the manganese-dependent superoxide dismutase gene (*sodA*), and the glyceraldehyde-3-phosphate dehydrogenase gene (*gap*) (Vázquez et al., [35]). The same assignments were inferred from the orthoANI values (Appendix A) and a phylogenomic analysis with type strains of the closest *Streptococcus* species carried out at the DSMZ Type Strain Genome Server (http://ggdc.dsmz.de/, accessed on 9 August 2022). Although closely related—and treated here as *S. equorum*—three isolates (5A3I, 11A1I, and 30A2I) showed dDDH and orthoANI values of 58.8–66.0% and 94.74–95.96%, respectively, with respect to the two *S. equorum* subspecies type strains. These values are lower than, or on the verge of, the most accepted thresholds (70% and 95% for dDDH and orthoANI, respectively), suggesting they may constitute a different taxonomic unit. The genetic diversity of the 13 sequenced strains was also examined by comparing their genome sequences with those in public databases. The results largely agreed with the high genetic diversity obtained by PCR-based genotyping. The scattered distribution of the database strains over the phylogenomic tree (Appendix A) strongly suggests the sequenced strains of this study are genetically unrelated.

### 2.3. Genome Analysis for ARGs

The genome sequences of the 13 strains were examined for genes involved in antimicrobial resistance by comparison with the sequences in the dedicated antimicrobial resistance databases CARD, NCBI-RGC, and ResFinder. Nucleotide sequences and deduced proteins of interest were further sought in PATRIC annotations and compared with sequences in the NCBI database by BLAST analysis. As a result, genes associated with resistance to six classes of antibiotics—penam (β-lactams), macrolides, lincosamides, phenicols, fluoroquinolones, and phosphonic acids (fosfomycin)—were detected (Table 2). Although this, the presence of well-known acquired genes associated with a concomitant phenotypic resistance was only found in a minority of the strains.

Despite all strains being susceptible to ampicillin and penicillin G, genes involved in the production of β-lactamase enzymes were possessed by all strains (Table 2). A *blaR1-blaZI* operon, which encodes a class A betalactamase (BlaZ, a penicillinase), was identified in five strains (5A3I, 11A1I, 30A2I, 48A3I and 50A2C). The operon was located in small contigs of 6.5–8.9 kbp within the same genetic context and surrounded by plasmid-associated genes. Electrotransformation of competent *S. aureus* cells with plasmid DNA from all five strains produced no viable ampicillin- or penicillin-resistant transformants. In addition to *blaZ*, CARD, NCBI-RGC, and ResFinder identified another gene in *S. equorum* 35A3C coding for another putative class A betalactamase: *bla*. Inspection of PATRIC annotations revealed *bla* to be present in all the studied genomes. The gene shared an identical genetic context in all strains and was flanked by chromosomally encoded genes. Nonetheless, the *bla* gene was polymorphic and the protein variants (Appendix A) may provide differential resistance.

Database searches detected an *mph*(C) gene in two strains (8A3C and 16A1C). Both strains also carried an *msr*(A) gene also found in five other strains (T17, 2A3C, 23A3C, 35A3C, and 50A2C). *mph*(C) and *msr*(A) code for, respectively, a macrolide 2’-phosphotransferase (inactivating the antibiotic) and an ABC-F type ribosomal protection protein (protecting the target). Nucleotide identity of the *mph*(C) gene led to a single Mph(C) protein, while Mrs(A) showed several variants, of which two (those of 8A3C and 16A1C, and of 50A2C) were associated with high erythromycin resistance (MIC 24–64 μg mL^−1^; Appendix A). All these strains, however, proved to be clindamycin-susceptible, even though two strains (1BCExtra and 2A3C) also carried an additional plasmid-encoded gene homologous to the *lnu*(A) gene from *Staphylococcus* spp. *lnu*(A) codes for a lincosamide nucleotidyltransferase thought to be involved in resistance to lincosamides. The presence of *lnu*(A) led to phenotypic examination for lincomycin resistance in all 13 strains. The MICs for this antibiotic varied widely (from 1 to 64 µg mL^−1^), with no correlation seen with respect to the presence or absence of *lnu*(A). Finally, although the MICs for ciprofloxacin were low, database searches identified a polymorphic *norA* gene in all strains (Appendix A). *norA* codes for an efflux pump of the major facilitator superfamily (MFS) and is known to be involved in fluoroquinolone resistance.

Although fosfomycin was not initially tested, a *fosB*/*fosD* polymorphic gene coding for a fosfomycin-inactivating enzyme was identified in the chromosome of all strains; the open reading frame (ORF) was disrupted in some of them. Phenotypic testing of the strains to fosfomycin led by this finding showed a surprisingly wide range of resistance (MICs from 2 to >1024 µg mL^−1^). Further, the strain showing the greatest resistance (1BCExtra) contained two copies of the gene, although one was disrupted. In fact, the complete copy of *fosB*/*fosD* in 1BCExtra (1BCExtra-1) was identified by the database searches to be an acquired gene. The copy was deemed to be chromosomally encoded, but it was flanked on one end by Tn*552*-associated ORFs. Further, the protein variant of strain 1BCExtra proved to be the most dissimilar to all others (Appendix A).

In contrast to the complex situation of most antibiotics, a *cat* gene matching phenotypic resistance to chloramphenicol was detected in strain 35A3C. Genes supporting a moderate phenotypic resistance to chloramphenicol (MIC 8–16 µg mL^−1^) in most strains were not identified. The *cat* gene encodes a type A-7 chloramphenicol *o*-acetyltransferase. It was located in a contig of 4.6 kbp harboring genes (*repB*, *mobCAB*) and signals (*dso*, *sso*, *oriT*) involved in plasmid replication and transfer (Appendix A). The Cat protein was identical to that encoded on plasmid pC221 and very similar to that of pC223, both plasmids from *S. aureus* (Appendix A). Further, the contig and the whole pC223 (GenBank accession no. NC_005243.1) were very similar at the nucleotide level (92% identity and 98% length coverage). By transforming the plasmid complement of 35A3C into *S. aureus* cells, chloramphenicol-resistant colonies were obtained. All transformants analyzed contained a plasmid with a restriction enzyme digestion profile compatible with that determined in silico for the *cat* contig. These results indicate that the contig contained the complete sequence of the *cat* plasmid of 35A3C (pCAT). The MIC for chloramphenicol in *S. aureus* increased from 4 µg mL^−1^ in the recipient to 32 µg mL^−1^ in the pCAT-containing transformants. In contrast, transformants were never recovered when plasmid DNA of 35A3C was electroporated into *Lactococcus lactis*, *Enterococcus faecalis*, or *Escherichia coli* cells, suggesting pCAT does not replicate in these hosts.

Finally, the use of antibiotic resistance databases also identified several genes (*qacJ*, *norC*, *sepA*, *sdrM*, and *qacJ*) coding for multidrug efflux transporters similar to proteins involved in resistance to disinfectants and antiseptics. Of these, *qacJ* from CL10P returned identity and coverage percentage ranges to genes in databases of 98.13–100% and 70.14–95.54%, respectively. *qacJ* was located in a small contig of (2323 bp) containing an ORF coding for a replication protein with a nucleotide identity of 93% to that in *S. aureus* plasmid pNVH01 (NC_004562.1).

## 3. Discussion

*S. equorum* populations can become large in a number of fermented foods [22,23,24], yet the antibiotic resistance properties of this species’ strains have been little studied [17,18,19,20,21]. In this work, strains of *S. equorum* isolated from a Spanish traditional blue-veined cheese were subjected to phenotypic, genetic, and genomic analyses in order to determine their antibiotic R/S profiles, to identify the genetic basis of the phenotypic resistances noted, and to search their genomes for antibiotic resistance-associated genes. As in other bacterial populations from cheese [36,37,38], the combined fingerprinting profiles revealed wide genetic diversity, which was confirmed by phylogenomic analysis of the sequenced strains. The examination of unrelated, non-clonal strains is important if the results of R/S assays are to be reliable and extendible to new *S. equorum* strains.

### 3.1. Phenotypic Testing and Proposal of S. equorum-Specific Cut-Offs

In agreement with most previous studies [17,18,19,20,21], low MICs were returned for most antibiotics in the present work; indeed, all were considered susceptible. Some MICs, however, fell above the cut-offs for *Staphylococcus* spp. established by EUCAST [33], although they were below those established by CLSI [34]; these strains were, therefore, also considered susceptible to the antibiotics concerned. Despite this, to exclude the presence of low-resistance conferring systems [39], genetic analyses of the strains showing the highest MICs might still be advisable. The MICs above the CLSI cut-off to penicillin G correlated well with an enhanced MIC for ampicillin, although a cut-off for this latter antibiotic has yet to be established. EUCAST has not developed cut-offs for ampicillin and penicillin G due to widespread resistance to these antibiotics among staphylococcal isolates, which makes their clinical use impractical [33]. The MIC values for ampicillin and penicillin G obtained with the microdilution assays were not reproduced by the MTS system. However, the largest MICs were always exhibited by the same strains, suggesting a differential resistance as compared to more susceptible ones. This resistance level, however, was much lower than that recorded in clinically resistant *S. aureus* strains (16–256 µg mL^−1^) [40]. Low resistance might be provided by dissimilar cell structures (cell-wall permeability, membrane charges, penicillin-binding protein variants, etc.) or the differential activity of unspecific transporters (multidrug systems, efflux pumps) [41]. The bimodal topology of the distribution curves for these antibiotics (Appendix A) strongly suggests that a part of the population has acquired resistance. The presence of colonies within the inhibition halos points towards undefined mutations accounting for such resistance. In contrast to β-lactams, some strains showed different (and quite high) MIC values for erythromycin and chloramphenicol. These values were considered compatible with the presence of dedicated resistance systems and the involvement of acquired (and possibly transferable) genes [15].

In the absence of species-specific R/S cut-offs, authorized agencies recommend the use of those of closely related species [33,42]. However, knowing whether a given strain is susceptible or resistant to an antibiotic is still challenging, particularly if the cut-offs cited by different sources (such as those of EUCAST and CLSI) are dissimilar [33,34]. In addition to clinical R/S cut-offs, microbiological (MCOFFs) or ecological cut-offs (ECOFFs) may also serve to distinguish susceptible from resistant strains [43]. MCOFFs and ECOFFs describe the MIC above which bacterial isolates may show phenotypically detectable acquired resistance mechanisms. This should better allow the identification of strains carrying transferable resistances, and thus help prevent their use in food (and feed) systems [42]. Examining the data in the literature has the potential to help establish *S. equorum* MCOFFs or ECOFFs, but differences between surveys in terms of the antibiotics tested, the concentration ranges contemplated, the methodologies used (disk diffusion, plate, and broth microdilution, Etest, etc.), and the media and culture conditions imposed, etc. [17,18,19,20,21], hamper any direct comparisons of results. Analysis of the distribution of MICs from large sets of antibiotic resistance assays using programs such as ECOFFinder [44] or NRI [45] was utilized in this study to propose *S. equorum*-specific cut-offs. The incorporation of new MIC data by surveying more strains from different sources can help establish robust and reliable cut-offs for this species.

### 3.2. Genome Analysis for Antibiotic Resistance Genes

Commensal and beneficial food-borne bacteria may play a role in disseminating antibiotic resistance to pathogens by acting as reservoirs of genetic determinants [3]. A comprehensive understanding of the whole pool of genes available and the transfer process is essential if we are to combat this resistance [6,15]. Genome sequencing and comparative analysis are currently the gold standards for characterizing the genetic potential of microorganisms, including the detection of ARGs in bacteria [46], the prediction of phenotypic resistance [47], and the assessment of its transferability [48]. Comparing the genomes of antibiotic-resistant and susceptible bacteria would further allow distinctions to be made between genes known to be spread through bacterial species and housekeeping genes involved in antibiotic resistance [49]. The distribution of the resistome in chromosomal and plasmid contigs has recently been reported following genome analysis of different *Staphylococcus* species [50,51], including *S. equorum* strains [52]. Chromosomal genes are found on large contigs harboring well-recognized housekeeping genes, while genes on plasmids are usually found in small contigs and in the vicinity of genes coding for replication proteins or proteins involved in plasmid maintenance and mobilization. In practical terms, the genome analysis such as that undertaken in the present work may lead to further phenotypic testing for antibiotics that are not included in commercial testing panels. It also allowed for acquired but silent genes to be detected. In the absence of antibiotic pressure, the acquisition of tightly regulated genes (silent genes) that do not lead to the phenotypic expression of resistance have been abundantly described [10,53]; of these, some might easily activate under certain conditions [54]. Then, the transfer of silent genes to clinically important pathogens could still lead to therapy failure [55].

A *cat* gene encoding a type A-7 chloramphenicol *o*-acetyltransferase was identified in one strain. Small plasmids conferring chloramphenicol resistance via *cat* genes coding for chloramphenicol *o*-acetyl transferases have been abundantly characterized in *Staphylococcus* species of different origins [56,57].

In the *S. equorum* strains of this work, only genes of the *bla* family were identified: an identical *blaR1-blaZI* operon on plasmid contigs in five strains, and a *bla* gene in all other strains on the bacterial chromosome. Whether the contigs harbor the complete plasmid molecules is not currently known. Neither *blaZ* nor *bla*, however, afforded moderate or strong resistance to ampicillin or penicillin G. Indeed, the strains carrying the genes for both betalactamases were among the most susceptible to both antibiotics. These results lead us to foresee that the phenotypic resistance to penicillin G is not due to acquired genes. Even though large plasmids have been recently introduced in Gram-positive bacteria by electroporation [58], no colonies were obtained by transforming plasmid DNA from the five strains into *S. aureus*, suggesting the *S. equorum* plasmidic BlaZ system is not functional. As reported for *S. aureus*, sequence variation in the *blaR1-blaZI* region [59], or mutations in the promoter region [60], may also account for the discrepancies observed between genotype and phenotype. As pointed out above, despite being silent, acquired genes may still represent a hazard.

The *msr*(A) and *fosB*/*fosD* genes proved to be polymorphic. The protein variants encoded by the different ORFs were thought to be responsible for the different MICs recorded for erythromycin and fosfomycin. As regards erythromycin resistance, the simultaneous presence of *msr*(A) and *mph*(C) in a single strain, as might be the case in some of the present strains, has previously been associated with enhanced MICs [61]. *norA* encodes an efflux pump of the major facilitator superfamily that has been associated with resistance to fluoroquinolones in *Staphylococcus* [62]. However, this gene in the *S. equorum* strains of this study was not associated with ciprofloxacin resistance. Lüthje and Schwarz [63] propose *norA* to be a core gene in staphylococci and suggest fusaric acid and siderophores be its natural substrates.

### 3.3. Transferability of Antibiotic Resistance Genes

Plasmids are the major route of dissemination of resistance determinants; therefore, resistance genes harbored on plasmids are the most likely to be transferred [64]. Plasmids such as pC221 and pC223 belong to the rolling-circle replicating (RCR) plasmids of the pT181 family and have been shown to replicate in several Gram-positive species, including staphylococci and *Bacillus subtilis* [65]. In addition to *cat*, *blaR1-blaZI*, and *lnu*(A) determinants were also found to be plasmid-encoded in this work. The in vitro conjugal transfer of *lnuA*-containing plasmids, and their lincomycin-associated resistance, from *S. equorum* to *Staphylococcus* spp., *E. faecalis* and *Tetragenococcus halophilus*, has already been reported [66]. *lnuA*-containing plasmids from the *S. equorum* cheese strains are larger (contigs of 32–34 kbp) than those in *S. equorum* strains from fermented seafood (2.6–2.8 kbp) [66], which might hinder their transfer. Differences in the size and mode of replication can influence the transfer and host range of plasmids, thus limiting their spread. In the present work, pCAT was successfully transferred to *S. aureus* by electroporation. The presence of an *oriT* sequence followed by ORFs encoding Mob proteins further suggests pCAT has a capacity for mobilization by other means (e.g., conjugation). Certainly, the 1BCExtra-1 *fosB*/*fosD* gene, which is thought to lie on the chromosome, was flanked by ORFs encoding Tn*552*-associated proteins, strongly suggesting horizontal acquisition by transposon-mediated integration. Indeed, Tn*552* has long been recognized for its capability to integrate plasmids on the chromosome [67]. Although posing a lesser risk than those in plasmids, this chromosomal fosfomycin resistance gene is thought to be easily transmitted via horizontal transfer.

## 4. Materials and Methods

### 4.1. Bacterial Strains and Culture Conditions

A total of 53 *S. equorum* isolates that had been recovered from ripened samples of traditional blue-veined Cabrales cheese and identified by partial amplification and sequencing of the gene encoding the 16S rRNA gene [35] were investigated in this work. Isolates were routinely cultured in Brain Heart Infusion (BHI) broth (VWR International, Darmstadt, Germany) supplemented with 2.5% (*w*/*v*) NaCl (3% in total), at 32 °C for 48 h. *Staphylococcus aureus* RN4220 was grown without agitation at 37 °C in Tryptone Soy Broth (TSB; Scharlab, Barcelona, Spain). *Enterococcus faecalis* 52c and *Lactococcus lactis* NZ9000 were grown at 32 °C in M17 broth (Formedium, Swaffham, UK) supplemented with 1% (*w*/*v*) glucose (GM17). *Escherichia coli* DH10B was grown in 2xTY broth at 37 °C with shaking. When required, agar (2% *w*/*v*) was added to the broth media for the preparation of solid plates.

### 4.2. Typing of the Strains

Total genomic DNA was isolated using the GenElute Bacterial Genomic DNA Kit (Sigma-Aldrich, St. Louis, CA, USA) following the manufacturer’s recommendations for Gram-positive bacteria. Genotyping of the strains was performed by combining the fingerprinting profiles obtained with primers BoxA2R (5′-ACGTGGTTTGAAGAGATTTTCG-3′), OPA18 (5′-AGGTGACCGT-3′) and M13 (5′-GAGGGTGGCGGTTCT-3′), as previously reported [36]. Clustering of the profiles was performed using the Unweighted Pair Group Method with Arithmetic Means algorithm (UPGMA) and Jaccard similarity coefficients, using MVSP v.3.21 software (Kovach Computing Services, Pentraeth, UK).

### 4.3. Antibiotic Testing

The MICs for 16 antibiotics in *S. equorum* strains was determined by broth microdilution using Sensititre EULACBI1 and EULACBI2 plates (Trek Diagnostic Systems, East Grinstead, UK). Briefly, colonies grown for 48 h on Mueller–Hinton (M-H) (Oxoid, Basingstoke, UK) agar plates were used to prepare cell suspensions in 0.9% NaCl solution (density corresponding to McFarland standard 1, i.e., ~3 × 10^8^ cfu mL^−1^). The suspensions were then further diluted 1:750 in M-H to achieve a final concentration of about 4 × 10^5^ cfu mL^−1^. One hundred microliters were then inoculated into the wells of the Sensititre plates and incubated at 32 °C for 48 h. MICs for ampicillin, penicillin G, and fosfomycin (Sigma-Aldrich), and lincomycin (ThermoFisher, Waltham, MA, USA) were also determined by microdilution in M-H broth using two-fold antibiotic dilutions. MICs were defined as the lowest antibiotic concentration at which no growth was observed.

When the concentration range of an antibiotic on the Sensititre plates was unable to pinpoint the actual MIC, and when MIC values had to be confirmed, MIC Test Strips (MTS; Liofilchem, Roseto degli Abruzzi, Italy) were used. This involved dipping a sterile cotton swab into a cell suspension corresponding to McFarland standard 0.5 (~1.5 × 10^8^ cfu mL^−1^) and using it to inoculate the surface of an M-H agar plate. After drying for 15 min, the antibiotic-containing strip was placed on the plate and incubated at 32 °C for 48 h. MICs for ampicillin and penicillin G were further confirmed with MTS.

Strains were considered susceptible or resistant based on the breakpoints recommended for *Staphylococcus* spp. in 2023 by either the European Committee on Antimicrobial Susceptibility Testing [33] or the American Clinical and Laboratory Standards Institute [34]. A strain was classified as resistant when the MIC exceed the breakpoint in more than one dilution.

### 4.4. MIC Analysis and Tentative R/S Cut-Offs

To suggest some tentative cut-offs, the distribution of the presently determined and literature-reported MICs was analyzed by methods based on statistics involving non-linear regression curves using ECOFFinder software version 2.1 (https://clsi.org/meetings/microbiology/ecoffinder/; accessed on 7 June 2022), as reported by Turnidge et al. [44], and the Normalized Resistance Interpretation (NRI) spreadsheet 2019 version (http://www.bioscand.se/nri/; accessed on 7 June 2022), as reported by Kronvall [45].

### 4.5. Whole-Genome Sequencing and Analysis

Genomic DNA from overnight cultures of *S. equorum* was extracted using the QIAmp DNA Mini Kit (Qiagen, Hilden, Germany) following the manufacturer’s protocol. High throughput sequencing libraries were prepared using standard methods and paired-end (2 × 150 bp) sequenced in a NovaSeq 6000 sequencer at GATC (Eurofins, Ebersberg, Germany). Genomes were annotated using PATRIC services (https://www.patricbrc.org/; accessed on 7 June 2022). For this, reads were first checked for quality with FastQC software version 0.11.9, and assembled in contigs using the Unicycler program (https://bio.tools/unicycler; accessed on 7 June 2022) after comparing different variables using the Quality Assessment Tool for Genome Assemblies (QUAST) (https://quast.sourceforge.net). Errors were polished using Pilon and Racon software (https://pkgs.org/). Annotation with PATRIC was performed using the RAST tool kit (RASTtk v1.073). Antibiotic resistance was investigated by genome analysis and comparison against sequences in the CARD database version 3.2.5 (https://card.mcmaster.ca/) (80% identity, 70% length coverage), the NCBI AMR Reference Gene Catalogue (using AMRFinderPlus: https://www.ncbi.nlm.nih.gov/pathogens/antimicrobial-resistance/AMRFinder/; accessed on 20 October 2022), and the ResFinder database (https://cge.cbs.dtu.dk/services/ResFinder/; accessed on 10 August 2022) (80% protein identity, 60% length coverage). PATRIC annotation data were also examined to seek out ARGs. Using BLAST tools (https://blast.ncbi.nlm.nih.gov/Blast.cgi; accessed on 20 October 2022), DNA and deduced protein sequences of interest were individually compared (at the nucleotide and amino acid levels, respectively) against information in the NCBI database. Whole-genome sequence data were used to ascertain the phylogenetic relationships between the sequenced strains and the type strains of related staphylococcal species by means of digital DNA–DNA hybridization (dDDH) and orthologous average nucleotide identity (orthoANI), as reported by Meier-Kolthoff and Göker [68] and Yoon et al. [69], respectively.

### 4.6. Phylogenetic and Phylogenomic Analyses

A phylogenomic comparison of the sequenced strains with six selected *S. equorum* strains from the NCBI database was carried out at the BV-BRC Server (https://www.bv.brc.org; accessed on 23 January 2023) using 500 genes. The phylogenetic tree for Cat proteins was constructed using sequences retrieved from the NCBI database (https://www.ncbi.nlm.nih.gov/protein/; accessed on 29 May 2023) aligned with MUSCLE and using the maximum likelihood estimation test. Alignment and phylogenetic reconstructions of other proteins were performed using the function “build” of ETE3 3.1.2 as implemented on GenomeNet (https://www.genome.jp/tools/ete/; accessed on 6 April 2023).

### 4.7. Isolation and Transformation of Plasmid DNA

Plasmid DNA from *S. equorum* strains was extracted and purified as described by O’Sullivan and Klaenhammer [70] with minor modifications. Briefly, 4 µL of mutanolysin (5 U µL^−1^), 5 µL of lysostaphin (2 mg mL^−1^), 20 µL of proteinase K (30 mg mL^−1^), and 4 µL of RNase (20 mg mL^−1^) were added to the lysis buffer. Electrocompetent cells of *E. coli*, *S. aureus*, *L. lactis*, and *E. faecalis* were prepared as reported by Sambrook and Russell [71], Schneewind and Missiakas [72], Holo and Nes [73], and Pérez et al. [74], respectively. Electrotransformation (electroporation) was performed using a Gene Pulser device (Bio-Rad, Richmond, CA, USA) following, as required, standard protocols for Gram-positive or Gram-negative bacteria. Transformants were selected on appropriate agar plates supplemented with antibiotics (chloramphenicol 10 μg mL^−1^; ampicillin and penicillin G 2, 4, and 8 μg mL^−1^), and, if recovered, analyzed to determine their plasmid content.

## 5. Conclusions

*S. equorum* strains from cheese phenotypically showed little clear-cut resistance to the tested antibiotics, and the concomitant presence of phenotypic resistance and supporting genes was only observed in a minority of strains. Acquired resistance by mutations was thought to provide low resistance to ampicillin and penicillin G and high resistance to erythromycin. Acquired genes—some of which were silent—known to be spread across diverse bacterial groups, involved in resistance to β-lactams, chloramphenicol, and fosfomycin, were found in the genome of some strains. Multi-resistant strains or strains carrying more than one acquired gene were not detected. Expressed or silent, ARGs in food bacteria are considered a hazard. Strains showing genes on large contigs containing well-recognized chromosomally encoded genes were identified. Others were found in smaller contigs and in the vicinity of genes coding for plasmid-replication proteins or proteins involved in plasmid maintenance or mobilization. These were deemed to be plasmid-encoded and thought to pose the greatest risk of horizontal transfer. Indeed, both the origin of replication of pCAT and its associated *cat* gene were found to be functional in *S. aureus*. Altogether, the above results suggest *S. equorum* is already a reservoir of acquired and transferable ARGs. Therefore, full phenotypic and genomic characterization of candidate strains to be used as starters or adjunct cultures in food fermentations is advisable.

## Figures and Tables

**Table 1 ijms-24-11657-t001:** Distribution of the Minimum Inhibitory Concentration (MIC) values of 16 antibiotics to 30 *Staphylococcus equorum* strains isolated from cheese obtained using a commercial microdilution test (Sensititre; Trek Diagnostic Systems).

Antibiotics	Number of Isolates with a MIC Value (µg mL^−1^)														*Staphylococcus* spp. Cut-Offs ^a^					*S. equorum* Cut-Offs
	0.03	0.06	0.12	0.25	0.5	1	2	4	8	16	32	64	128	256	EUCAST		CLSI			This Work ^b^
															S (≤)	R (>)	S (≤)	I (=)	R (≥)	R (≥)
Gentamicin					30 ^c^										2	2	4	8	16	1
Kanamycin							30 ^c^								8	8	(-)			4
Streptomycin					12 ^c^	16	2								(-)		(-)			4
Neomycin			30 ^c^												(-)		(-)			0.25
Tetracycline				14	14	2									1	1	4	8	16	2
Erythromycin				3	14	6	**3**	**1**			**2 ^d^**	**1 ^d^**			1	1	0.5	1–4	8	2
Clindamycin			2	8	**7**	**7**	**5**	**1**							0.25	0.25	0.5	1–2	4	4
Chloramphenicol								3	23	3		**1**			(-)		8	16	32	32
Ampicillin	4 ^c^	8	8	3	2 ^d^	3 ^d^	2 ^d^								(-)		(-)			0.5
Penicillin G	7 ^c^	4	12			**2 ^d^**	**4 ^d^**	**1 ^d^**							(-)		0.12	(-)	0.25	0.5
Vancomycin					21	9									4	4	4	8–16	32	2
Quinupristin-dalfopristin				2	11	16	**1**								1	1	1	2	4	4
Linezolid						5	19	6							4	4	4	(-)	8	8
Trimethoprim					2	10	8	10							4	4	8	(-)	16	8
Ciprofloxacin				21 ^c^	9										0.001	1	(-)	2	4	1
Rifampicin			27 ^c^	**3**											0.06	0.06	1	2	4	0.5

^a^ Cut-offs (in µg mL^−1^) established by the European Committee on Antimicrobial Susceptibility Testing (EUCAST) [33] and the Clinical and Laboratory Standards Institute (CLSI) [34]. Output: S, susceptible; I, intermediate; R, resistant; (-), breakpoint not established. In grey and in bold, strains with an MIC value considered resistant by EUCAST, CLSI or both. ^b^ The cut-offs proposed included the results of this work and those in Marty et al. [18]. ^c^ Strains that do not grow on the lowest concentration of the antibiotic assayed (MIC ≤ to this value). ^d^ Current MIC established with the MIC Test Strip system (MTS; Liofilchem).

**Table 2 ijms-24-11657-t002:** Genes related to antibiotic resistance identified in the genome of the *Staphylococcus equorum* strains isolated from cheese of this study.

Antibiotic Class/Gene	Activity/Resistance Mechanism	Strain(s)	Identified by Database and/or Pipeline	% Identity/% Length Coverage ^a^	Amino Acid (aa) Identity/Total aa	Location ^b^ (Size kbp)	Maximum Homology to Protein
Penams							
*blaR1-blaZI*	Class A beta-lactamase/antibiotic inactivation (AI)	5A3I, 11A1I, 30A2I, 48A3I, 50A2C	CARD, NCBI-RGC, PATRIC, ResFinder	100/100	281/281	Plasmid (6.50–8.90)	WP_069819195.1
*bla*	Class A beta-lactamase	CL10P, 1BCExtra, 5A3I, 8A3C,16A1C, 50A2C	Manual revision	99–100/100	279–282/282	C	WP_002508531.1
		T17		99/100	279/282		WP_064783177.1
		2A3C, 11A1I, 30A2I		99/100	282/282		WP_069813561.1
		23A3C		100/100	282/282		WP_119627547.1
		35A3C	CARD, NCBI-RGC, ResFinder	100/100	282/282		WP_046465027.1
		48A3I	Manual revision	99/100	280/282		WP_197911012.1
Macrolides							
*mph*(C)	Macrolide 2’-phosphotransferase/AI	8A3C, 16A1C	CARD, NCBI-RGC, ResFinder	100/100	299/299	C	WP_119544566.1
*msr*(A)	ABC-F type ribosomal protection protein/target protection	T17, 2A3C, 23A3C, 35A3C	PATRIC, ResFinder	99/100	488/488	C	WP_069813611.1
		8A3C, 16A1C		99/100	487/488		WP_046465994.1
		50A2C		100/100	488/488		WP_069854570.1
Lincosamides							
*lnu*(A)	Lincosamide nucleotidyltransferase/AI	1BCExtra, 2A3C	Manual revision	100/100	161/161	Plasmid (32.0–34.60)	WP_069813868.1
Phenicols							
*cat*	Type A chloramphenicol *o*-acetyl transferase/AI	35A3C	CARD, NCBI-RGC, PATRIC, ResFinder	100/100	215/215	Plasmid (4.6)	WP_053038759.1
Fluoroquinolones							
*norA*		CLP10, 1BCExtra, 48A3I	PATRIC	99/100	385/386	C	WP_002508336.1
	Major facilitator superfamily of efflux pumps/antibiotic secretion	T17, 23A3C,35A3C		100/100	386/386		WP_064783100.1
		2A3C, 8A3C, 16A1C, 50A2C	PATRIC, ResFinder	100/100	386/386		WP_021339414.1
		5A3I *, 11A1I, 30A2I	PATRIC	99 *–100/100	385 *–386/386		WP_069832674.1
Phosphonic acids							
*fosB*/*fosD*	Fosfomycin bacillithiol transferase/AI	1BCExtra-1	CARD, NCBI-RGC, PATRIC, ResFinder	100/100	139/139	C	WP_000616116.1
		T17, 23A3C, 35A3C	PATRIC	100/100	139/139		WP_031266123.1
		1BCExtra-2 *	CARD, PATRIC	84/51	70/139		WP_056935383.1
		2A3C *,8A3C *, 16A1C *, 50A2C *	CARD	84/32	45/139		WP_031266123.1
		5A3I, 11A1I, 30A2I	CARD, NCBI-RGC, PATRIC, ResFinder	100/100	139/139		WP_069833353.1
		48A31 *	CARD, PATRIC	83/51	70/139		WP_031266123.1

^a^ Identity and coverage of DNA or protein sequence, depending on the database. Only identity and length coverage percentages of proteins are stated. ^b^ C, chromosome; P, plasmid. Plasmid location was considered when genes encoding plasmid-replicating, mobilization and/or maintenance proteins were found in the same contig. The size referrers to the size of the contig; that of the *cat* gene contained the whole plasmid. * Disrupted genes containing premature stop codons.

## Data Availability

The complete genome sequences of the 13 *S. equorum* strains examined were deposited in the GenBank database under the BioProject and BioSample accession numbers PRJNA940711, and SAMN33577425 through SAMN33577437.

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
