# Peer review of "Antibiotic Resistance/Susceptibility Profiles of Staphylococcus equorum Strains from Cheese, and Genome Analysis for Antibiotic Resistance Genes"

_ijms, 2023, doi:10.3390/ijms241411657_

Round 1
Reviewer 1 Report
1. In line 160, “was” is preceded by the subject, “in” needs to be removed.
2. In lines 100 and 401, for the genus name “Staphylococcus”, please use italicized form.
3. The experiment conducted by the author is very comprehensive, but it would be better to use methods such as RT qPCR to observe the expression of these resistance genes under antibiotic stress
Minor editing of English language required
Author Response
Reviewer 1
- In line 160, “was” is preceded by the subject, “in” needs to be removed.
The organization of the sentence might be rather odd, which may lead to a misunderstanding. For clarity, the sentence has now been reorganized as follows:
Although this, the presence of well-known acquired genes associated with a concomitant phenotypic resistance was only found in a minority of the strains.
- In lines 100 and 401, for the genus name “Staphylococcus”, please use italicized form.
Thank you for the comment; the scientific names have been all checked and italicized.
- The experiment conducted by the author is very comprehensive, but it would be better to use methods such as RT qPCR to observe the expression of these resistance genes under antibiotic stress.
We sincerely acknowledged the suggestion of the reviewer. We believe this might be the subject of subsequent works. This study, however, aimed to assess the phenotypic antibiotic resistance profiles of a set of S. equorum strains, a species in which antibiotic resistance testing has been scarce. In addition, we also aimed to correlate antibiotic resistance with genetic traits, and we succeeded at least in two cases: the strains resistant to chloramphenicol, in which an active and transferable cat gene was found, and the strain resistant to high levels of fosfomycin, in which an acquired variant of fosB/fosD was detected. Expression of the genes under antibiotic stress will undoubtedly provide valuable information on the role of such genes in the survival of antibiotic resistance-containing strains. The manuscript, which includes different aspects of the antibiotic resistance problem, is already too long; this part might be addressed in the future.
Reviewer 2 Report
The title of the manuscript is good. English language has good quality. Tables have acceptable quality. Some sections of the manuscript need some changes.
1. Please rewrite the part "abstract" according to order below:
+ Brief Information about Antibiotic resistance/susceptibility
+ The importance of Staphylococcus equorum
+ the importance of genome analysis for antibiotic resistance genes of Staphylococcus equorum
+ brif data about material and method
+ brief data about results and conclusion
2. Line 33-35 in page 1
Please rewrite the sentence "Their promise has, however, been tarnished by the appearance of resistance and the subsequent transfer of the associated
genetic determinants to pathogenic bacteria, hindering the treatment of infections [2]." this sentence is a little hard to understand. Please make it more simple.
3. All over the section "Introduction" there are some multipple references. Please reform all of them
4. Line 55 in page 2
The sentence "Staphylococcus equorum is a member of the coagulase-negative ..." needs a logical and proper connection with its previous paragraph. Please make this connection in order to keep continuity of the text.
5. In the part "Results" line 118-120
This sentence belongs to the part "material and methods" pmease reconsider it.
Note: In the part "Results", please write only and only about "Results". Any other extra
information (including the purpose of performing a special method, any comparison with the results of other surveys and ...) should be written in its proper section.
6. Line 163-165 and lime 190-192 in part "Results"
This sentence has reference and belongs to the section "Discussion". Please refer to note in comment 5 and reconsider this sentence.
7. About the part "Discussion"
Please rewrite this part according to notes below:
First: categorize all of your results based on their importance (from the most important one to the least important)
Second: after that, turn each one of your results into some subheadings
Third: after that, discuss about them one by one
Forth: make comparisons between your
results and the results of other similar and relevant surveys
7. Please check and adjust the "Reference list" based on the regulations of reference list of journal. (Titles, doi, the name of journal and ... )
Author Response
Reviewer 2
The title of the manuscript is good. English language has good quality. Tables have acceptable quality. Some sections of the manuscript need some changes.
- Please rewrite the part "abstract" according to order below:
+ Brief Information about Antibiotic resistance/susceptibility
+ The importance of Staphylococcus equorum
+ the importance of genome analysis for antibiotic resistance genes of Staphylococcus equorum
+ brif data about material and method
+ brief data about results and conclusion
We mostly agree with the reviewer to include short introductory sentences in the Abstract. We already had these in our drafts. However, these were deleted in order not to exceed the suggested 200 words, because we thought that the text included cannot be shortened without losing essential information. Should the journal allow us to include such introductory parts, we´ll be very happy to modify the Abstract in this way (see below).
The only aspect that we do not think is necessary is that about the importance of genome analysis in the study of the genetic basis of antibiotic resistance. Genome analysis is pivotal in the characterization of most, in not all, microbiological traits, and, in this manuscript, is sufficiently acknowledged in the Discussion section.
We also believe all other aspects mentioned by the reviewer had already been met: phenotypic testing by microdilution and genome are both mentioned as the way to assess the phenotypic resistance and the genetic basis of the resistances, respectively. The last sentence of the Abstract is already considered a short conclusion.
In food, bacteria carrying antibiotic-resistance genes could play a prominent role in the spread of resistance. Staphylococcus equorum populations can become large in a number of fermented foods, yet the antibiotic-resistance properties of this species have been little studied.
- Line 33-35 in page 1
Please rewrite the sentence "Their promise has, however, been tarnished by the appearance of resistance and the subsequent transfer of the associated genetic determinants to pathogenic bacteria, hindering the treatment of infections [2]." this sentence is a little hard to understand. Please make it more simple.
Thank you for your comment. English usage of all our manuscripts is reviewed by Adrian Burton (https://physicalevidence.es/english/about-us/), a freelance writer with long expertise in scientific communication. Although he is very thorough and professional, sometimes his sentences are too literary, which makes them difficult to read and understand. We try to soften as much as possible such sentences, but occasionally some sentences may escape our attention. The sentence has been rewritten now, as follows:
The initial success, however, has been tarnished by the appearance of resistance and the subsequent transfer of the associated genetic determinants to pathogenic bacteria, hindering the treatment of infections.
- All over the section "Introduction" there are some multiple references. Please reform all of them.
That´s a quite good observation; thank you. Because multiple aspects are being touched on in the manuscript (phenotypic testing, various resistance genes, plasmids, transference, etc.), the number of resulting references is too large. The whole manuscript has now been revised for non-relevant or redundant references, and, where possible, the most appropriate one has been retained.
- Line 55 in page 2
The sentence "Staphylococcus equorum is a member of the coagulase-negative ..." needs a logical and proper connection with its previous paragraph. Please make this connection in order to keep continuity of the text.
The paragraph has been reordered to give such continuity to the text. The last two sentences have now been advanced to the start of the paragraph and modified to fit this position as follows:
Staphylococcus equorum is a species for which R/S cut-offs have yet to be developed. Indeed, work on antibiotic resistance in S. equorum strains has been particularly scant, and involved mostly disk diffusion assays [24, 37-39], which do not allow the establishment of reliable cut-offs. S. equorum is a member of the coagulase-negative…
- In the part "Results" line 118-120
This sentence belongs to the part "material and methods" please reconsider it.
Note: In the part "Results", please write only and only about "Results". Any other extra information (including the purpose of performing a special method, any comparison with the results of other surveys and ...) should be written in its proper section.
Sorry for the repetition of sentences from the Material and Methods section in the Results. Occasionally, short introductory sentences might be adequate, but in most others are simply redundant, which is the case here. The sentence has now been deleted in the revised version.
- Line 163-165 and lime 190-192 in part "Results"
This sentence has reference and belongs to the section "Discussion". Please refer to note in comment 5 and reconsider this sentence.
The two sentences have been amended as suggested by the reviewer, and the accompanying sentences slightly modified for clarity.
A blaR1-blaZI operon, which encodes a class A betalactamase (BlaZ, a penicillinase), was identified in five strains (5A3I, 11A1I, 30A2I, 48A3I and 50A2C).
The presence of lnu(A), which codes for an efflux pump of the major facilitator superfamily (MFS) known to be involved in fluoroquinolone resistance, led to phenotypic…
- About the part "Discussion"
Please rewrite this part according to notes below:
First: categorize all of your results based on their importance (from the most important one to the least important)
Second: after that, turn each one of your results into some subheadings
Third: after that, discuss about them one by one
Forth: make comparisons between your results and the results of other similar and relevant surveys.
Subheadings, which were already considered and (finally) discarded, have now been added to the Discussion section. Data within each of the paragraphs have already been addressed one by one, categorized by their (of course our subjective) importance, and discussed and compared with data from the literature. Unfortunately, few works dealing with antibiotic resistance and genome analysis in S. equorum are available in the literature. Therefore, most comparisons in the Discussion have inevitably dealt with data from the reference species in the genus, S. aureus.
Nonetheless, some paragraphs have still been reordered, such as that appearing the first in the previous manuscript that has been moved to the Genome analysis section. And a new one discussing some aspects of the cat gene has also been included under this heading, which was only marginally commented on under the transferability section in the previous version. Minor small amendments were also done to the Conclusion sections, where the last sentence of the last sentence of the Discussion was moved to avoid a kind of repetition.
- Please check and adjust the "Reference list" based on the regulations of reference list of journal. (Titles, doi, the name of journal and ... )
The final reference list has been fully revised to comply with the IJMS´ Instructions for Authors. Hope they all meet the journal´s policy and layout.
Reviewer 3 Report
The manuscript described the 16 antibiotic R/S profile of Staphylococcus equorum isolated (n=30) from cheese products. The study used conventional microbiological culture methods and genomic technology to support the study objective.
Overall, it is a well-written and well-organized manuscript, and the study has scientific merit. A list of minor comments and suggestions is below for the authors to take into consideration while revising the manuscript.
1. Line 17-18: The sentence “However, the genetic data did not always correlate with the phenotype” should be moved after line 25 (….norA in all stains) as a takeaway message of the results revealed in this study.
2. Line 58: the shift of example as part of the study objective from dairy to meat products needs a better contextual presentation.
3. Line 74-76: This sentence is somewhat confusing. Please consider revising or fragmenting.
Section 4.1: The method of strain isolation needs a detailed description.
4. Please consider replacing next-generation sequencing (NGS) with highthroughput sequencing (HTS) throughout the manuscript.
5. Section 4.6: The manuscript should include the Staphylococcus DSMZ strain number and relevant details. Was it Staphylococcus equorum?
6. Why 16S RNA gene was not used for strain identification?
7. Provide references for the using ECOFFinder and NRI worksheet programs.
8. Provide the year and version of the EUCAST and CLSI protocols.
9. Define how the current study can be applied in the dairy industry for routinely monitoring the antibiotic R/S S. equorum population. It is unrealistic for the industry to isolate and identify and use biochemical and genomic technology to determine the ab R/S stains of S. equorum in dairy products. Moreover, ab(R) S. equorum strain may transfer the resistance genes to other Staphylococcus and non-staphylococcus strains in real samples. How would the current study go about profiling the microbiota?
10. Define the real applicability of this study in context to the last sentence of the Abstract “The antibiotic R/S status and gene content of S. equorum strains intended to be employed in food systems should be carefully determined.”
Author Response
Reviewer 3
The manuscript described the 16 antibiotic R/S profile of Staphylococcus equorum isolated (n=30) from cheese products. The study used conventional microbiological culture methods and genomic technology to support the study objective.
Overall, it is a well-written and well-organized manuscript, and the study has scientific merit. A list of minor comments and suggestions is below for the authors to take into consideration while revising the manuscript.
- Line 17-18: The sentence “However, the genetic data did not always correlate with the phenotype” should be moved after line 25 (….norA in all stains) as a takeaway message of the results revealed in this study.
Well appreciated, thank you. However, we think the sentence can be just in front of the description of those genes found that do not correlate with a phenotypic resistance (at the end of line 21 in the previous version).
- Line 58: the shift of example as part of the study objective from dairy to meat products needs a better contextual presentation.
That´s right; introducing meat seems to be odd under the dairy background of the manuscript´s content. The sentence has been modified to state that S. equorum contributes to the sensory properties of some fermented foods, without any reference to meet. It now appears as:
Enzyme activities of CNS species contribute to the development of flavours and other sensorial characteristics of fermented foods [30, 31].
- Line 74-76: This sentence is somewhat confusing. Please consider revising or fragmenting.
The sentence has been split into two sentences for a better understanding, as follows:
This prompted further phenotypic testing with respect to additional antibiotics, such as lincomycin and fosfomycin. Genome analysis also allowed acquired resistance caused by mutations to be distinguished from resistance afforded by acquired genes.
Section 4.1: The method of strain isolation needs a detailed description.
In this work, we started with a collection of 53 S. equorum isolates that had already been isolated and identified in previous works, although the results are yet unpublished. The sentence has now been modified to state this fact, as follows:
A total of 53 S. equorum isolates that had been recovered from ripened samples of traditional blue-veined Cabrales cheese and identified by partial amplification and sequencing of the gene encoding the 16S rRNA gene (Vázquez et al., unpublished), were investigated in this work.
- Please consider replacing next-generation sequencing (NGS) with highthroughput sequencing (HTS) throughout the manuscript.
The term next generation sequencing was present only once in the manuscript. The expression was replaced by high throughput sequencing as suggested.
High throughput sequencing libraries were prepared using standard methods and…
- Section 4.6: The manuscript should include the Staphylococcus DSMZ strain number and relevant details. Was it Staphylococcus equorum?
The accession numbers in the DSMZ collection are included in the corresponding figure; these type species are automatically selected by the DSMZ Type Strain Genome Server. The confusion might come from the fact that we provided two independent phylogenomic analyses, and perhaps this was not well explained in the section. For the sake of clarity, and because the only phylogenomic analysis that matters in this study is the second one (the unrelatedness of the strains tested and sequenced), the first phylogenomic analysis will be skipped, and the genomic identification of the 13 sequenced strains, which gave identical results as those obtained in the previous study, summarized as data not shown.
A phylogenomic comparison of the sequenced strains with six selected S. equorum strains from the NCBI database was carried out at the BV-BRC Server (https://www.bv.brc.org) using 500 genes.
The same assignments were inferred from the orthoANI values (Table S2) and a phylogenomic analysis with type strains of the closest Streptococcus species carried out at the DSMZ Type Strain Genome Server (http://ggdc.dsmz.de/) (data not shown).
- Why 16S RNA gene was not used for strain identification?
16S rRNA amplification and sequencing was initially used for the identification of all isolates. This is now clearly stated in the response to your previous comment (see above). However, when the genome sequence is available, genome analysis is considered to give a more precise identification of the strains at species, subspecies, and strain levels. As pointed out in the manuscript, although highly related to S. equorum, three isolates might belong to a newly non-described Staphylococcus species.
- Provide references for the using ECOFFinder and NRI worksheet programs.
The methods have already been referenced, although the references were well apart from the programs, which might have caused some confusion. The paragraph has now been modified to locate each of the references close by their corresponding worksheet programs, as follows:
To suggest some tentative cut-offs, the distribution of the presently determined and literature-reported MICs was analysed by methods based on statistics involving non-linear regression curves using ECOFFinder software (https://clsi.org/meetings/microbiology/ecoffinder/), as reported by Turnidge et al. [61], and the Normalized Resistance Interpretation (NRI) spreadsheet (http://www.bioscand.se/nri/), as reported by Kronvall [62].
- Provide the year and version of the EUCAST and CLSI protocols.
The EUCAST and CLSI protocols were the latest available; both Agencies released new versions this year. This was, and still is, clearly stated in their respective references. To make it clear, the paragraph was slightly modified to include the year in the text, as follows:
Strains were considered susceptible or resistant based on the breakpoints recommended for Staphylococcus spp. in 2023 by the European Committee on Antimicrobial Susceptibility Testing [40] and by the American Clinical and Laboratory Standards Institute [41]. A strain was classified as resistant when the MIC exceed the breakpoint in more than one dilution.
- Define how the current study can be applied in the dairy industry for routinely monitoring the antibiotic R/S S. equorum population. It is unrealistic for the industry to isolate and identify and use biochemical and genomic technology to determine the ab R/S stains of S. equorum in dairy products. Moreover, ab(R) S. equorum strain may transfer the resistance genes to other Staphylococcus and non-staphylococcus strains in real samples. How would the current study go about profiling the microbiota?
Thanks a lot for this suggestion; we believe it is a good one. We are aware that some of these analyses (particularly genome sequencing and analysis) cannot be implemented in routine analysis in the industry (except perhaps for big starter companies, which have already done so for all their commercial strains). The study can also serve to suggest the phenotypic testing to which the S. equorum isolates intended to be used as starters in foods should be subjected. This might include the testing, for instance, for fosfomycin and lincomycin resistance, antibiotics that at the moment are not included in the commercial testing systems. A short sentence in the Discussion has been modified to emphasize this fact.
In practical terms, genome analysis such as that undertaken in the present work may lead to further phenotypic testing for antibiotics that are not included in commercial testing panels.
- Define the real applicability of this study in context to the last sentence of the Abstract “The antibiotic R/S status and gene content of S. equorum strains intended to be employed in food systems should be carefully determined.”
This comment is very much related to the previous one. The use of large numbers of antibiotic-resistant free strains should preclude the development of strains carrying antibiotic-resistant genes (thus avoiding subsequent transfer of genes to other members of the microbiota). This is a message for researchers involved in the identification of starter candidates and to the starter companies selecting robust strains among the candidates to be produced in industrial settings and released to the food manufacturing sector. We do believe this is well explained in the last sentence of the conclusion and it does not need further details.
Altogether, the above results suggest S. equorum is already a reservoir of acquired and transferable ARGs. Therefore, full phenotypic and genetic characterization of candidate strains to be used as starters or adjunct cultures in food fermentations is advisable.
Round 2
Reviewer 2 Report
All of my comments are considered. Thanks